# A Stress Syndrome Prototype Reflects Type 3 Diabetes and Ischemic Stroke Risk: The SABPA Study

**DOI:** 10.3390/biology10020162

**Published:** 2021-02-18

**Authors:** Leoné Malan, Mark Hamer, Roland von Känel, Roelof D. van Wyk, Anne E. Sumner, Peter M. Nilsson, Gavin W. Lambert, Hendrik S. Steyn, Casper J. Badenhorst, Nico T. Malan

**Affiliations:** 1Hypertension in Africa Research Team (HART), North-West University, Potchefstroom 2520, South Africa; Roland.VonKaenel@usz.ch (R.v.K.); nico.t.malan@gmail.com (N.T.M.); 2Division of Surgery & Interventional Science, Faculty of Medical Sciences, University College London, London WC1E 6BT, UK; m.hamer@ucl.ac.uk; 3Department of Consultation-Liaison Psychiatry and Psychosomatic Medicine, University Hospital Zurich, 8091 Zurich, Switzerland; 4Surgical Ophthalmologist, 85 Peter Mokaba Street, Potchefstroom 2531, South Africa; rdvisu@gmail.com; 5Section on Ethnicity and Health, Diabetes, Endocrinology and Obesity Branch, National Institute of Diabetes and Digestive and Kidney Diseases, Bethesda, MD 20892, USA; anne.sumner@nih.gov; 6National Institute of Minority Health and Health Disparities, National Institutes of Health, Bethesda, MD 20892, USA; 7Department of Clinical Sciences, Lund University, SE-205 02 Malmö, Sweden; peter.nilsson@med.lu.se; 8Iverson Health Innovation Research Institute, Swinburne University of Technology, Hawthorn, VIC 3122, Australia; glambert@swin.edu.au; 9Baker Heart & Diabetes Institute, Melbourne, VIC 3004, Australia; 10Statistical Consultation Services, North-West University, Potchefstroom 2520, South Africa; faans.steyn@gmail.com; 11Anglo American Corporate Services, Sustainable Development Department, Johannesburg 2017, South Africa; dr.cas.badenhorst@gmail.com

**Keywords:** stress, type 3 diabetes, neuron-specific enolase, S100B, retina, glaucoma, neurodegeneration

## Abstract

**Simple Summary:**

The relationship between diabetes, dementia and neuronal injury defines type 3 diabetes (T3D). Similar retinal vessel changes were observed in Alzheimer’s and chronic stressed individuals. The aim was to assess associations of chronic stress with T3D and retina pathology. An established validated chronic stress and stroke risk score (hereafter Stress) stratified participants into Stressed (high stress) and non-Stressed (low stress) groups. Early dementia signs (insulin sensitivity, cognitive dysfunction, neuronal injury) were evident in Stressed compared to non-Stressed individuals. Stress was associated with poor retinal oxygen perfusion, potentiating glaucoma risk. Stress was further related to four dementia risk markers (cognitive dysfunction; telomere shortening, neuronal injury and waist circumference), which comprised a novel Stress syndrome prototype. This prototype showed high risk for T3D as well as retinal vein widening and perfusion deficits, indicating stroke risk. Findings should increase awareness on the detrimental role of chronic stress in the human brain.

**Abstract:**

Type 3 diabetes (T3D) accurately reflects that dementia, e.g., Alzheimer’s disease, represents insulin resistance and neurodegeneration in the brain. Similar retinal microvascular changes were observed in Alzheimer’s and chronic stressed individuals. Hence, we aimed to show that chronic stress relates to T3D dementia signs and retinopathy, ultimately comprising a Stress syndrome prototype reflecting risk for T3D and stroke. A chronic stress and stroke risk phenotype (Stressed) score, independent of age, race or gender, was applied to stratify participants (N = 264; aged 44 ± 9 years) into high stress risk (Stressed, N = 159) and low stress risk (non-Stressed, N = 105) groups. We determined insulin resistance using the homeostatic model assessment (HOMA-IR), which is interchangeable with T3D, and dementia risk markers (cognitive executive functioning (cognitive_exe-func_); telomere length; waist circumference (WC), neuronal glia injury; neuron-specific enolase/NSE, S100B). Retinopathy was determined in the mydriatic eye. The Stressed group had greater incidence of HOMA-IR in the upper quartile (≥5), larger WC, poorer cognitive_exe-func_ control, shorter telomeres, consistently raised neuronal glia injury, fewer retinal arteries, narrower arteries, wider veins and a larger optic cup/disc ratio (C/D) compared to the non-Stressed group. Furthermore, of the stroke risk markers, arterial narrowing was related to glaucoma risk with a greater C/D, whilst retinal vein widening was related to HOMA-IR, poor cognitive_exe-func_ control and neuronal glia injury (Adjusted R^2^ 0.30; *p* ≤ 0.05). These associations were not evident in the non-Stressed group. Logistic regression associations between the Stressed phenotype and four dementia risk markers (cognitive_exe-func_, telomere length, NSE and WC) comprised a Stress syndrome prototype (area under the curve 0.80; sensitivity/specificity 85%/58%; *p* ≤ 0.001). The Stress syndrome prototype reflected risk for HOMA-IR (odds ratio (OR) 7.72) and retinal glia ischemia (OR 1.27) and vein widening (OR 1.03). The Stressed phenotype was associated with neuronal glia injury and retinal ischemia, potentiating glaucoma risk. The detrimental effect of chronic stress exemplified a Stress syndrome prototype reflecting risk for type 3 diabetes, neurodegeneration and ischemic stroke.

## 1. Introduction

One of the major modifiable risk factors for stroke is diabetes, which may cause pathologic changes in cerebral blood vessels [1,2]. Impairments in cerebral glucose utilization and insulin resistance [3] further accompany the initial stages of cognitive impairment [4] and neurodegenerative diseases [5]. Neergaard and colleagues [6] also showed that subjects above the threshold of the homeostatic model assessment of insulin resistance (HOMA-IR > 2.6) had 47% greater odds of cognitive dysfunction.

Most insulin in the brain is derived from circulating pancreatic insulin and enters the brain primarily via selective transport across the capillary endothelial cells of the blood–brain barrier [1] and the blood–retinal barrier (BRB) [7]. Peripheral insulin resistance (IR) is accompanied by central nervous system IR and might increase the risk of dementia neuropathology [8]. Indeed, IR at cellular level in midlife was shown to be an independent risk factor for brain amyloid accumulation in individuals without dementia [8]. Using the term type 3 diabetes (T3D) rather accurately reflects the fact that dementia, e.g., Alzheimer’s disease (AD), represents insulin resistance in the brain, with molecular and biochemical features that overlap with both type 1 diabetes mellitus [9] and type 2 diabetes [9,10,11]. Type 1 diabetes mellitus, or insulin deficiency due to autoimmune ß-cell destruction, and type 2 diabetes, due to a progressive loss of ß-cell secretion sufficient to overcome insulin resistance [10], have not, however, been related to AD [9]. The relationship between T3D and AD has been based on the fact that both the processing and clearance of amyloid-β (Aβ) are attributed to impaired insulin signaling, and that IR mediates the dysregulation of bioenergetics and progress to AD [11,12]. Determining progression of T3D neurodegeneration towards early AD is challenging, and chronic stress might provide more insight as a potential risk factor in this process.

The brain and the retina share similar blood barrier physiology and neuronal glia functioning. Stroke risk markers, i.e., retinal arterial narrowing and vein widening, might further explain a link between T3D and retinopathy or neurodegeneration [7,13]. Therefore, assessment of peripheral biochemical markers indicative of neuronal glia injury can complement neuropsychiatric and diabetic retinopathy evaluation [14,15,16,17,18,19,20]. Neuron-specific enolase (NSE) [16] and cytosolic calcium-binding protein, S100B [17], have been associated with preclinical diabetes diagnosis and prognosis of optic nerve diseases [16,17,18]; as well as S100B-related ischemic conditions in the Sympathetic activity and Ambulatory Blood Pressure in Africans (SABPA) cohort [14]. S100B is concentrated in astrocytes of retinal neural layers and released when retinal neuronal damage is apparent [17]. Indeed, Vujosevic and co-workers [18] administered S100B intravitreal injections that induced ischemia and glaucoma-like symptoms with early optic nerve axon degeneration, followed by retinal cell body damage. Contradicting findings regarding NSE levels were, however, shown, as high levels of NSE, a glycolytic enzyme, were found in the serum of diabetic retinopathy patients, whereas lower levels (2.00–7.50 ng/mL) were associated with brain atrophy [16] and the metabolic syndrome [20]. Glucose dysregulation was furthermore related to dementia signs, namely DNA damage (shorter telomeres) [21], inflammation and platelet aggregation [22].

Interestingly, however, a majority of cognitive neurodegenerative disease patients did not carry risk genes [4,21], which raises the question of whether the initial trigger indicates environmental influences. This notion is supported by Benarroch [23], as most glucose uptake in neurons occurs via insulin-regulated glucose transporters 3 (GLUT3) and 4 (GLUT4) in the emotional limbic cortical and subcortical regions, which have a high density of insulin receptors. It is, then, also to be expected that insulin will affect psychiatric functioning, as it was among the earliest drug treatments for psychiatric disorders [24]. Determining stress-related T3D risk markers is critically needed, as similar retinal neurovascular dysregulation response patterns have been observed in individuals with chronic stress from the SABPA study [14,15] and in Alzheimer’s dementia individuals [25].

We therefore aimed to assess whether (a) chronic stress will relate to more T3D signs and retinopathy, and whether (b) chronic stress will relate to dementia risk markers as a Stress syndrome prototype, reflecting T3D neurodegeneration and stroke risk.

## 2. Materials and Methods

### 2.1. Design and Participants

The Sympathetic activity and Ambulatory Blood Pressure in Africans (SABPA) prospective study was used as the data source for the present analysis. Methods are well-described elsewhere [26]. Urban-dwelling Black and White teachers with similar socio-economic status and who had access to medical aid benefits were under investigation. A complete dataset was compiled of individuals participating at both baseline and the 3-year follow-up phase, and seasonal changes were avoided (Figure 1).

Exclusion criteria at baseline were pregnancy, lactation, tympanum temperature ≥37.5 °C, the use of psychotropic substances or α- and β-blockers and blood donors or individuals vaccinated within 3 months prior to data collection.

### 2.2. Protocol

A 24-h ambulatory blood pressure and ECG monitoring apparatus, the Cardiotens CE120^®^ (Meditech, Budapest, Hungary), was fitted between 07:00 and 09:00 on working days (Monday–Thursday) at the teachers’ schools (Appendix A). The 24-h ECG data were used to determine time domain heart rate variability (HRV) (Appendix A). A standardized 24-h diet commenced and participants resumed their normal school and extra-curricular activities. Thereafter, they were transported to the North-West University for extensive clinical assessments to be performed under well-controlled conditions [26]. Each participant received his/her own private bedroom for an overnight stay at the North-West University guest house facilities. Demographic and General Health questionnaires were completed. Participants took a last beverage before 22:00 (coffee/tea and biscuits) and were asked to fast from 22:00 to 07:00. The next morning at 07:00, anthropometric measures were taken, followed by carotid intima media sonar scanning to determine bifurcation stenosis (Appendix A). Blood sampling was performed and physical activity measures proceeded before being transported to school. Feedback reports and referral letters to doctors were given to each participant within 7 days.

#### 2.2.1. Fundus Imaging and Retinopathy

Retinal vessel imaging considering arteriolar and venular calibers (hereafter referred to as arteries and veins) was performed at 3-year follow-up (Appendix A). Food and alcohol intake, smoking and exercise were prohibited one hour prior to measurements. Fundus imaging history and voluntary consent were obtained after participants were introduced to the procedure. A trained registered nurse screened participants for acute anterior angle chamber glaucoma risk using a small light source. Mydriasis was induced in the right eye of 262 (99.2%) individuals by means of a drop containing tropicamide, 1%, and benzalkonium chloride 0.01% (*m*/*v*). Fundus imaging was performed with a retinal vessel analyzer with a Zeiss FF450^Plus^ camera and the software VesselMap 2, Version 3.02 (Imedos Systems GmbH, Jena, Germany) (Appendix A), which automatically determined the retinal artery and vein count. Retinal vessel calibers were measured as monochrome images by manually selecting first-order vessel branches in a measuring zone located between 0.5 and 1.0 optic disc diameters from the margin or the optic disc. After selection of the vessel, software automatically delineated the vessels’ measuring area. The Knudtson formula was used to determine estimates from the 6 largest arteries and veins. As the image scale of each eye was unknown, the values of the retinal arteries and veins were expressed as measuring units (MU). One measuring unit is equivalent to 1 micrometer when the dimensions of the eye being examined correspond to those of the normal Gullstrand eye. Reproducibility was computed for a randomly selected cohort with a correlation coefficient of 0.84.

#### 2.2.2. Diastolic Ocular Perfusion Pressure

Diastolic ocular perfusion pressure was measured to indicate hypo-perfusion risk. Hypo-perfusion was evaluated after instilling a local anesthetic drop (Novasine Wander 0.4% Novartis) and measuring intra-ocular pressure (IOP) with the Tono-Pen Avia Applanation Tonometer (Reichert 7-0908, ISO 9001, New York, NY, USA). Right-eye diastolic ocular perfusion pressure (mmHg) was calculated (office diastolic blood pressure minus intra-ocular pressure). Clinical observations for diabetic and hypertensive retinopathy were evaluated on color stereo optic disc photographs by an experienced and qualified ophthalmologist blinded to the group. Evaluations included, e.g., determination of the optic cup/disc ratio, arterio-venous nicking (AV nicking) and focal narrowing. AV nicking indicates indentation of retinal veins by stiff (arteriosclerotic) retinal arteries as a sign of chronic hypertension and vascular dysregulation [27]. Focal narrowing of retinal arteries increases with progressive glaucomatous optic neuropathy and may include thinning of the neuroretinal rim area when optic nerve head damage is apparent [28].

#### 2.2.3. Cognitive Executive Functioning Control

Accumulating evidence suggests that cognitive executive functioning control or the Stroop test [29] elicits attentional control mechanisms as an early marker for the transition from healthy cognitive aging to early-stage dementia of the Alzheimer type in the frontotemporal cortex areas [30]. For the current investigation, we used the Stroop test as an early dementia risk marker to indicate cognitive dysfunction [30,31]. The Stroop test (Appendix A) assesses attentional processing of simultaneously occurring sensory information in the context of selective attention, cognitive set shifting and response inhibition in the prefrontal cortex. A template containing five words printed in highly distinguishable colors (“blue”, “green”, “red”, “yellow”) in random order but written in incongruent colors was shown to participants (Appendix A). The ink color of a given word had to be identified verbally and individuals were encouraged to progress as fast as possible within 1 min and were corrected when wrong answers were given. Prior to the test, they were informed that they will receive a monetary incentive according to their performance on completion of the test, which served as motivation to improve performance. An interference score was calculated that represents the number of correct answers produced during the fixed period of 1 min. A lower score indicates that the individual found it more difficult to inhibit interference. Two trained scientists (registered nurse and MD) were involved in supervision of the Stroop test and scoring of all teachers at baseline. Perception of the stressfulness of the Stroop test was assessed on a 7-point Likert scale (Appendix A).

#### 2.2.4. Anthropometry and Physical Activity Measures

Waist circumference (WC) as marker of central obesity was measured in triplicate by registered level II anthropometrists (N = 2) and the mean was used for statistical analyses. A non-extensible flexible anthropometric tape was used to measure WC at the midpoint between the lower costal rib and the iliac crest, perpendicular to the long axis of the trunk, and not at the narrowest point, for standardization purposes. The intra- and inter-variability was less than 5%. Mean total energy expenditure over 7 days was derived from an Actiheart accelerometer (GB06/67703; CamNtech Ltd., Upper Pendrill Court, Papworth Everard, Cambridgeshire CB233UY, UK).

#### 2.2.5. Biochemical Analyses

Participants were in a semi-recumbent position for at least 30 min before blood sampling in both study phases. A registered nurse obtained all fasting blood samples before 09:00 from the antebrachial vein branches of the dominant arm of each participant using a winged infusion set. All blood samples were obtained from never-thawed serum/plasma/citrate samples, handled according to standardized procedures and stored at −80 °C until analyses.

The following dementia-related risk markers indicating cardiometabolic perturbations [1,2,3,4,5,6], neurodegeneration [1,2,3,4,5,6,7] and vascular dysregulation [14,15,27] were specified and included in the current investigation (in no particular order): poorer cognitive executive functioning control [30]; neuronal glia injury (increased S100B and NSE) [19]; increased central obesity or waist circumference (WC) [32]; endothelial dysfuntion (increased von Willebrand factor (VWF)) [33]; depressed time domain heart rate variability (HRV) (Appendix A) [34]; increased insulin resistance/HOMA-IR [8] and inflammation (C-reactive protein/CRP) [35]; and shorter telomeres [21].

Sodium fluoride glucose, serum levels of insulin, triglycerides, total cholesterol:high-density lipoprotein, cotinine (nicotine metabolite as indicator of smoking habits [36], gamma-glutamyl-transferase/GGT (as marker of alcohol consumption [37] and ultra-sensitive C-reactive protein (CRP) were analyzed with an enzyme rated method (Unicel DXC 800-Beckman and Coulter, 4300 N. Harbor Blvd., Fullerton, CA 92835 U.S.A), homogeneous immunoassay (Modular ROCHE Automized systems, Basel, Switzerland) and a particle-enhanced turbidimetric assay (Cobas Integra 400 plus, Roche, Basel, Switzerland), respectively. A dysregulated hypothalamic–pituitary–adrenal cortex axis (HPA axis) contributes to cognitive decline and dysfunctional growth hormones [38], e.g., serum total insulin-like growth factor 1 (IGF-1) and insulin-like growth factor binding protein 3 (IGFBP-3). IGF-1 and IGFBP-3 further protect nerve cells against neurodegenerative processes [39] and were determined using immunoradiometric assays from Immunotech, Beckman Coulter, Brea, California: Catalogue no. A15729 (inter-assay % CV 4.49; intra-assay % CV 2.92) and Catalogue no. DSL-6600 (inter-assay % CV 3.2–9.3; and intra-assay % CV 2.71–7.95), respectively. Whole-blood EDTA glycated hemoglobin (HbA_1C_) was analyzed with turbidimetric inhibition immunoassays (Cobas Integra 400 Plus, ROCHE Basel, Switzerland).

The American Diabetes Association guidelines were used to define pre-diabetes (HbA_1C_ ≥ 5.7%) and diabetes status (HbA_1C_ ≥ 6.5%) [10]. The homeostatic model assessment of insulin resistance (HOMA-IR) was determined using the following formula: fasting glucose × fasting insulin/405. The median (interquartile range) HOMA-IR result was 2.9 (1.8–4.7), and Q4 was at least 4.7. HOMA estimates β-cell function and insulin sensitivity based only on fasting glucose and insulin concentrations and gives a representation of insulin secretion adjusted for insulin sensitivity [40]. Citrate VWF antigen concentration as a marker of endothelial dysfunction, coagulation activation and ischemic stroke risk [41] was measured with a “sandwich” ELISA assay. A polyclonal rabbit anti-VWF antibody and a rabbit anti-VWF-HRP antibody (DAKO, Bloemfontein, Free State, South Africa) were used to form the assay. The 6th International Standard for VWF/FVIII was used to set the standard curve against which the samples were measured. Serum NSE and S100B were analyzed with an electrochemiluminescence immunoassay (e411, ROCHE, Basel, Switzerland) with intra- and inter-assay coefficients of less than 5%. Mitochondrial DNA (mtDNA) variation utilizing the MutPred program in the SABPA study [42] was determined to establish an association between chronic stress and maternal lineage. To determine leukocyte telomere length (LTL), genomic DNA was extracted. Reference DNA samples were prepared, and all isolated DNAs were mixed together in equal proportions, representing the average of all analyzed SABPA patients (N = 255). All experimental DNA samples were assayed in triplicate and PCRs were performed with the CFX96 Touch™ Real-Time PCR Detection System (Bio-Rad, Hercules, CA, USA) in a 25-µL volume (Appendix A) and stored at −20 °C. HIV-positive status (N = 10) was considered in regression analyses (Appendix A).

### 2.3. Statistical Analyses

Statistica version 13.3 (TIBCO Software Inc., Palo Alto, Santa Clara, CA, USA, 2018) was used for data analyses. We proceeded from previous investigations in the SABPA study [14,15] by applying a validated chronic stress and stroke risk phenotype (hereafter Stressed) score, independent of age, race or gender (see Section 6). Participants were stratified into high stress risk (Stressed, N = 159) and low stress risk (non-Stressed, N = 105) groups. The phenotype biomarkers for the current investigation were not included in any statistical models. All variables with non-normal distributions were Box–Cox transformed except for age, cognitive_exe-func_, diastolic ocular perfusion pressure, artery count and bifurcation stenosis.

#### 2.3.1. Chronic Stress and Stroke Risk Phenotype Contribution to T3D and Retinopathy Risk

Independent *t*-tests were used to compare clinical characteristics of the Stressed vs. non-Stressed groups whilst comparisons of proportions and prevalence were tested by using chi-square (χ^2^) tests. Dementia risk markers were compared between Stressed vs. non-Stressed groups using single one-way ANCOVAs, adjusted for age. Mean changes over 3 years in dementia risk markers were calculated using dependent sample *t*-tests within the Stressed and non-Stressed groups. McNemar’s case–control tests were performed to compare incidence and recovery frequencies of HOMA-IR over 3 years within the Stressed and non-Stressed groups. Pearson correlations were determined between cognitive_exe-func_, dementia risk markers (telomere length, NSE, S100B, WC, HOMA-IR, VWF, CRP, standard deviation of normal-to-normal (NN) intervals (SDNN)) and perception of stressfulness of the Stroop test. Multiple linear regression analyses determined associations between retinal vessel calibers, dementia and retinopathy risk markers within the Stressed and non-Stressed groups at baseline and 3-year follow-up. Dependent variables in Models 1 and 2 were retinal artery and venous calibers obtained at 3-year follow-up. Independent variables included baseline age, dementia risk markers (cognitive_exe-func_, NSE, S100B, HOMA-IR, CRP) and dyslipidemia as well as retinal risk marker data obtained at follow-up (including diastolic ocular perfusion pressure, optic nerve cup-to-disc ratio, retinal artery count and carotid bifurcation stenosis). When the retinal artery was a dependent variable, the retinal venous diameter was included as a predictor and vice versa. The F value to enter in regression models was fixed at 2.5.

#### 2.3.2. Chronic Stress and Stroke Risk Phenotype-Related Dementia Risk Markers in the Devlopment of a Stress Syndrome Prototype

A logistic regression analysis was performed, and the chronic stress and stroke risk phenotype (Stressed) was used as a dichotomous dependent variable to develop a Stress syndrome prototype to maximize the clinical utility of stress risk. The prototype development was scientifically grounded and conducted empirically to increase clinical usefulness [43]. Baseline dementia risk markers (cognitive_exe-func_, Box–Cox transformed markers, telomere length, NSE, S100B, WC, VWF, CRP and time domain 24-h HRV) were used as continuous predictors. An additional logistic regression analysis was performed to determine the association between Stressed and Box–Cox transformed mitochondrial DNA MutPred load, as a marker of maternal lineage [42]. Only significant Stressed-related dementia risk marker associations were used to develop the novel Stress syndrome prototype score using a receiver operating characteristic (ROC) area under the curve analysis. The statistical significance level was set at *p* ≤ 0.05 (two-tailed). A final logistic regression analysis computation was used to determine probability of risk for the Stress syndrome by including continuous predictors, i.e., Box–Cox transformed insulin resistance (HOMA-IR), S100B, retinal artery and vein caliber/AV nicking, retinal artery count, optic nerve cup/disc ratio, carotid bifurcation stenosis, diastolic ocular perfusion pressure and dyslipidemia. The statistical significance level was set at *p* ≤ 0.05 (two-tailed).

Statistical significance level was set at *p* ≤ 0.05 (two-tailed).

## 3. Results

### 3.1. Clinical Characteristics

In total, 264 participants with a mean age of 45.3 ± 9.1 years were followed for a 3-year period. As shown in Table 1, men (72%) were more prone to be classified as having chronic stress (*p* < 0.05) compared to their female counterparts (28%). The Stressed group showed higher prevalence of smoking and greater alcohol consumption as well as more dementia and retinal vascular dysregulation risk signs compared to the non-Stressed group (*p* < 0.05). Compared to their counterparts, the Stressed group also presented higher diabetes-prone symptoms (higher prevalence of ethnicity-specific central obesity, dyslipidemia, low-grade inflammation, lower cognitive_exe-func_ control scores, lower IGF-1 and IGFBP-3 levels, higher glycated hemoglobin and insulin, increased prevalence of hyperinsulinemia, triglyceridemia, pre-diabetes, diabetes, HOMA-IR, hypertension (74%) as well as increased retinopathy prevalence (62%), fewer retinal arteries, increased arterial narrowing, vein widening and AV nicking (74%) and higher perfusion deficit levels (glia ischemia (S100B) and higher diastolic ocular perfusion pressure) as risk markers of stroke [14].

#### 3.1.1. Chronic Stress and Stroke Risk Phenotype Contribution to T3D and Retinopathy Risk

In Figure 2, some dementia risk marker values adjusted for age were adversely affected in the Stressed group compared to the non-Stressed group, *p* ≤ 0.001 (see unadjusted values in Table 1). As shown in Figure 2a, the Stressed group had poorer cognitive_exe-func_ control, increased glial injury (S100B), increased WC, VWF and trend for depressed time domain heart rate variability prevalence of 27% vs. 17% (SDNN ≤ 100 ms) (*p* = 0.06). In Figure 2b, it is shown that the Stressed group had higher HOMA-IR, shorter telomeres, more neuronal injury (NSE) and increased CRP values. In Appendix A (Appendix A), poor cognitive_exe-func_ control in the Stressed and non-Stressed groups was related to telomere shortening (*p* < 0.05). Cognitive_exe-func_ control was inversely related to perception of Stroop test stressfulness, S100B and VWF; and positively related to HOMA-IR and WC in the Stressed group only.

In Table 2, three-year changes in dementia risk markers with respect to the stress phenotype are presented. NSE increased (*p* ≤ 0.001), whereas S100B was consistently higher in the Stressed group. WC and VWF increased (*p* ≤ 0.001) in both the Stressed and non-Stressed groups over the 3-year period. Incidence frequency for upper-quartile HOMA-IR (25%) in the Stressed group was apparent with lower recovery frequency (3%). No 3-year frequency variance was evident for severe HOMA-IR in the non-Stressed group.

In Table 3, retinal arterial narrowing is linearly associated with hypo-perfusion (diastolic ocular perfusion pressure) and a larger optic nerve cup/disc ratio in the Stressed group. Venous widening was associated with poor cognitive_exe-func_ control, hypo-perfusion, low NSE, higher ischemia (S100B) and HOMA-IR. In the non-Stressed group, a higher count of retinal arteries was associated with wider arteries as well as narrower veins, indicating less retinopathy and stroke risk [14,15]. Retinal vessel caliber risk markers were not related to changes in dementia risk markers over the 3-year period. No associations were found between retinal and dementia risk markers in the non-Stressed group or after excluding HIV-positive status individuals in any group.

#### 3.1.2. Chronic Stress and Stroke Risk Phenotype-Related Dementia Risk Markers and Development of a Stress Syndrome Prototype

In Table 4, a logistic regression model is presented of the statistically significant dementia risk markers (cognitive_exe-func_, telomere, NSE and WC), which predicted the Stress risk phenotype. The logistic model including only these four significant Stress risk phenotype-related dementia risk markers fitted well (Hosmer and Lemeshow test, *p* = 0.94). No association was found between the Stress phenotype and mtDNA.

In Figure 3, the Stress syndrome prototype comprises of four Stressed-related dementia risk markers (cognitive_exe-func_; Box–Cox transformed telomere length; neuron-specific enolase/NSE and waist circumference). An area under the curve of 0.80 is displayed at an optimal cut-point of 0.52, with 85% sensitivity and 58% specificity.

To illustrate, a retinal image of a male participant (Figure 4), who was positively identified as being in the Stressed and Stress syndrome prototype groups, is presented.

The novel Stress syndrome risk score was applied hereafter (Table 5) and linearly associated with HOMA-IR (OR 7.72), glia ischemia (OR 1.27) and retinal vein widening (OR 1.03). HOMA-IR, as a marker of insulin sensitivity, showed strong clinical significance.

## 4. Discussion

In this study, we applied a validated chronic stress and stroke risk phenotype score to evaluate the important role of chronic stress in the pathogenesis of T3D and retinopathy risk. The main findings of our analysis showed that the Stressed group had (1) increased T3D signs (higher HOMA-IR, poor cognitive_exe-func_, neuronal glia injury, increased central obesity, VWF and inflammation, depressed HRV, shorter telomeres), retinal perfusion deficits and glaucoma risk; and (2) that the Stress risk phenotype and four related dementia risk signs (cognitive_exe-func_, telomere, NSE and WC) comprised a novel Stress syndrome prototype. The Stress syndrome prototype exemplified chronic stress as a contributing factor with higher odds for T3D neurodegenerative morbidity and ischemic stroke risk.

### 4.1. Chronic Stress and Stroke Risk Phenotype Contribution to T3D and Retinopathy Risk

Only 38% of the chronic Stressed group had a prevalence of severe HOMA-IR but poorer ability for cognitive interference inhibition control. This might suggest a risk for cognitive-related dementia [30] when chronically stressed. In support, Glück and co-workers [45] showed impairment in glucose regulation and associated poorer performance on the Stroop test, whereas Neergaard and colleagues [6] reported that subjects with HOMA-IR >2.6 had 47% greater odds of cognitive dysfunction. Indeed, insulin resistance occurred in the brains of individuals with cognitive-related neurodegenerative diseases, independent of concurrent type 2 diabetes [46]. Data from a Swedish study support this notion, where continuous inverse correlations were shown between glucose levels and cognitive test results also for people without diabetes [47]. However, impaired neuroplasticity itself might also account for metabolic risk and neurodegeneration via dysregulation in the HPA axis [16] as well as cognitive dysfunction in the prefrontal cortices and hippocampus [48]. In support, HPA axis hypo-activity reflected non-adaptation to stress in the SABPA cohort [15]. Currently the Stressed group reported higher perception of stressfulness indicating non-adaptation to the Stroop test and related poor performance.

Considering insulin receptors, it is known that glia insulin receptors are downregulated in response to chronically high levels of insulin, whereas neuronal insulin receptors are not [49]. This may complicate our understanding of the effects of insulin signaling on brain function and neuronal glia protection/injury in the brain. The lower IGF-1 and IGFBP-3 and higher HOMA-IR levels in the Stressed group might further support inactivation or downregulation of retinal glia insulin receptors and subsequent endothelial dysfunction or glaucoma risk [50]. Decreases in IGF-1 concentration, a neurotrophic peptide in the central nervous system, in the Stressed group might therefore be regarded as a potential biomarker for early diagnosis or even prognosis of impaired insulin signaling [37]. IGF-1 and insulin together with cytokines activate common signaling pathways necessary for the reprogramming of Müller glia cells and retinal redevelopment upon injury [51].

Presently, the risk of neuronal glia injury and more dementia risk signs in the Stressed group showed detrimental effects on retinal function. Retinopathy is one of the most common microvascular complications of diabetes and a major cause of vision impairment in working-age adults [52]. In our teachers’ cohort, the Stressed group had increased arterial narrowing and vein widening, both stroke risk signs and emerging endothelial dysfunction in the inner retinal neural layer or BRB [14,15]. Concurrent perfusion deficits in the Stressed group could alter the integrity of the BRB [53,54]. Specifically, perfusion deficits encompassing hypo-perfusion, neuronal glia injury (NSE increases and consistently raised S100B) alongside HOMA-IR [14,19] can induce this change. Arterial narrowing and associated larger cup/disc ratio were apparent in the Stressed group and may further increase the risk for glaucoma and optic nerve head atrophy, particularly when considering the fewer retinal arteries (or loss of arteries) and less neuronal glia protection observed in the vulnerable Stressed group. The increases in NSE [55] and resultant S100B release have previously been correlated with the extent of neuronal damage [19]. Higher S100B levels had a direct destroying impact on the axons of the optic nerve, and damage of the retinal cell bodies seems to be a consequence of this axon loss [17] as well as being a risk for cognitive dementia [55]. Once again, these findings were only observed in the Stressed group, underscoring the notion that chronic stress might trigger a detrimental cycle in the brain that impairs insulin signaling. The 3-year increases in serum NSE in this group might sensitively reflect injury of retinal neurovascular units as one of the main pathological mechanisms of early diabetic cognitive impairment [56].

### 4.2. Chronic Stress and Stroke Risk Phenotype-Related Dementia Risk Markers and Development of a Stress Syndrome Prototype

Clinical stress-related dementia risk markers (cognitive_exe-func_, telomere length, NSE and WC) comprised a novel Stress syndrome prototype. The Stress syndrome prototype was associated with impaired insulin sensitivity and retinopathy as indicators of neurodegeneration [4,27]. Each of the stress-related dementia risk markers has several vascular dysregulation implications, e.g., poor cognitive inhibition control was associated with impaired glucose regulation [44] as well as DNA damage or telomere shortening [1]. A U-shaped association was shown between telomere length and risk of Alzheimer’s in the general population [21]. It aligns well with our finding that shorter telomeres were associated with poor cognitive executive functioning control. T3D risk further increases with impaired insulin sensitivity, which had predictive value (odds ratio 7.3) for the Stress syndrome prototype, indicating high clinical significance. Retinopathy may also develop as a complication of chronic hyperglycemia and lead to increased oxidative stress and potential shortening of telomeres [57]. Inevitably, the role of DNA maternal lineage emerges. A significant role for mitochondrial DNA variation in association with either hypertension or hyperglycemia in the SABPA cohort could not be identified [42], nor currently with chronic stress. This supports the role of environmental rather than genetic factors contributing to chronic stress.

We further observed 3-year increases in NSE as a marker of neuronal injury, which increases the risk for diabetic retinopathy [16]. According to Haque et al. [16], increases in NSE might activate inflammatory cytokines, chemokines and other inflammatory mediators to initiate axonal damage. The consistent release of S100B from astrocytes in retinal neural layers [17] underscores the predictive risk (odds ratio 1.3) for retinal ischemic-related neuronal damage increases [14,19] in the Stress syndrome prototype group.

An inflammatory mediator, namely central obesity [51], has been associated with metabolic perturbations in the SABPA cohort [44] and other cohorts [58,59] as well as with global measures of brain atrophy and cognitive decline such as executive function, learning and memory in adults and children [60]. This is of particular importance in people of African descent [44,59], as central obesity has been identified as an independent risk factor for the Stress syndrome prototype. The Atherosclerosis Risk in Communities (ARIC) study showed that mid-life, but not late-life, vascular risk factors increase the risk of an amyloid-positive scan using positron emission tomography (PET) and that central obesity independently predicted an amyloid-positive PET scan [61]. Increased triglycerides, as a risk factor for metabolic syndrome in the Stressed syndrome prototype, might induce brain insulin receptor resistance to further reflect a) enhanced secretion of triglyceride lipoproteins and b) impaired clearance of these lipoproteins [5,62].

An alarming level of metabolic perturbations at baseline when chronically stressed was observed with prevalence of central obesity (66%), low-grade inflammation (54%), hypertriglyceridemia (35%), high alcohol consumption (42%) and severe HOMA-IR (38%), which may mask and complicate our understanding of dementia or T3D risk signs. Findings may surely underscore the role of stress as a trigger for these metabolic perturbations and impaired insulin sensitivity or type 3 diabetes.

While the prospective nature and the well-controlled protocol, executed within a cohort with similar socio-economic status, are strengths of the study, our investigation was limited by the sample size and should be replicated in larger cohorts. Measures for advanced amyloid function in the brain to confirm cognitive functioning or decline are advised. The severity of dementia was not ascertained by the Clinical Dementia Rating (CDR) scale, cognitive function by the Mini Mental State Examination (MMSE) or neuroimaging findings with magnetic resonance imaging/PET. We recommend that the Stroop score, as a generic defense response [63], concurring with chronic stress [64] may have dual advantages both as a stress test and as an early T3D dementia risk marker.

### 4.3. Translational Clinical Value

Chronic stress may play an important role in the pathogenesis of type 3 diabetes, neurodegenerative morbidity and ischemic stroke risk.

## 5. Conclusions

In conclusion, chronic stress facilitated less neuronal glia protection with neurovascular changes suggesting glaucoma risk as well as retinal ischemia risk. A novel stress syndrome prototype enabled detection of type 3 diabetes and ischemic stroke susceptibility in the brain–retina axis. It further emphasizes the detrimental role of chronic stress in brain insulin resistance and neurodegeneration.

## 6. Patents

An ePCT International Application was filed on 31 July 2020 (Application No.: PCT/IB2020/057269): Method of determining risk for chronic stress and stroke. The phenotype revealed good discriminatory ability with a positive prediction of stroke risk in a prospective cohort, independent of age, race or gender (area under the receiver operating characteristic curve: 0.82 (95% CI: 0.75–0.85); *p* ≤ 0.001 for a positive prediction with 85% sensitivity/58% specificity). The phenotype biomarkers were determined with standardized methods [15,26,64,65].

## Figures and Tables

**Figure 1 biology-10-00162-f001:**
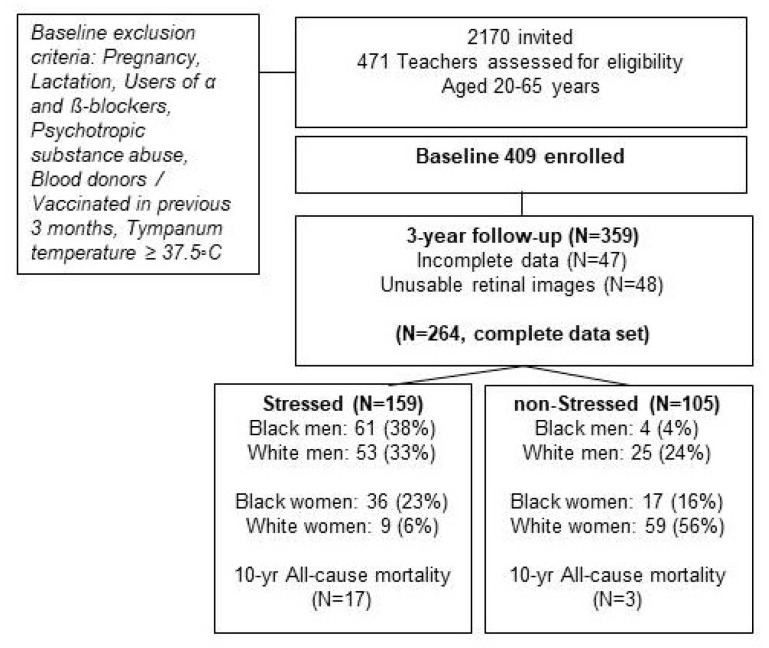
The Sympathetic activity and Ambulatory Blood Pressure in African (SABPA) prospective cohort stratified into Stressed and non-Stressed groups.

**Figure 2 biology-10-00162-f002:**
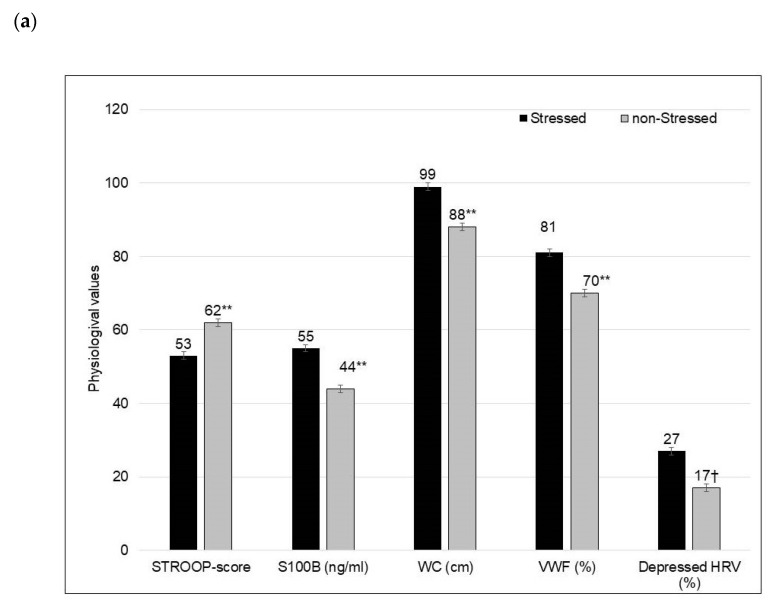
Comparing (mean ± SE) dementia-related risk markers adjusted for age in Stressed vs. non-Stressed groups. (**a**) Comparison of Stroop scores, S100B/glia injury, WC and VWF values and time domain 24-h HRV prevalence. (**b**) Comparison of HOMA-IR, telomere length, NSE and CRP values. Definitions: STROOP, cognitive_exe-func_ control; S100B, glia injury; WC, waist circumference; VWF, von Willebrand factor; Depressed HRV, prevalence of depressed time domain 24-h heart rate variability (SDNN ≤ 100 ms); SDNN, standard deviation of the normal-to-normal (NN) intervals between adjacent QRS complexes which equal the square root of variance; NSE, neuron-specific enolase or neuronal injury; CRP, C-reactive protein. **, *p* ≤ 0.001; †, *p* = 0.06.

**Figure 3 biology-10-00162-f003:**
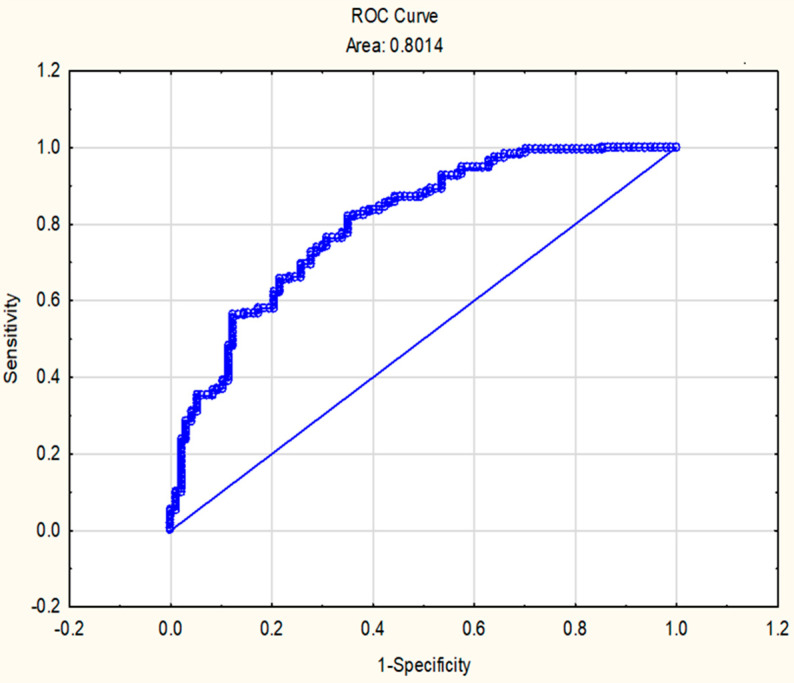
Stressed-related dementia risk markers comprised a novel Stress syndrome cut-point of 0.52 in a South African cohort (N = 236), using a receiver operating characteristic (ROC) curve. The area under the curve (AUC) (95% CI) was 0.80 (sensitivity 85%; specificity 58%).

**Figure 4 biology-10-00162-f004:**
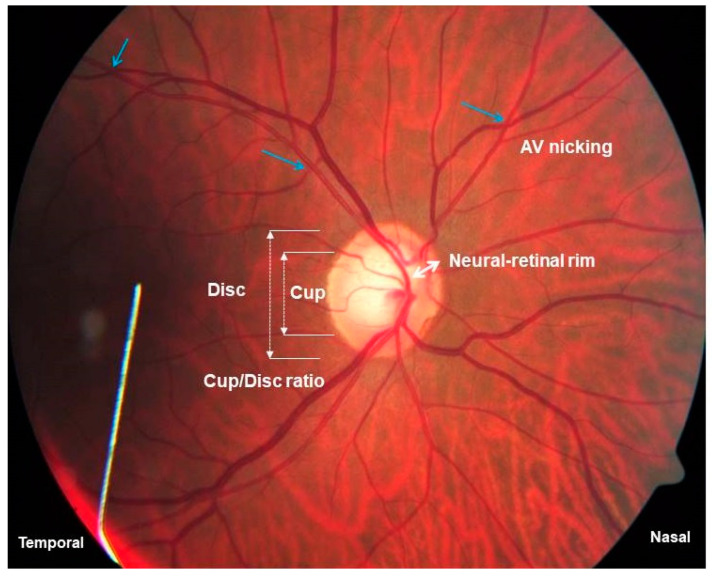
Presenting the retinal image of a male participant, fulfilling criteria for being in the Stressed and Stress syndrome prototype groups; and where male dominance was pertinent at 72% and 66%, respectively. The medians (lower–upper quartile ranges) of dementia and retinopathy risk marker values are reported: lower cognitive executive functioning control, 55 (45–65); central obesity, 105 cm (84–104 cm); shorter telomeres, 0.9 ng/µL (0.8–1.1 ng/µL); higher NSE, 12.7 ng/mL (7.2–11.8 ng/mL); raised S100B, 0.5 µg/L (0.03–0.06 µg/L); larger optic nerve cup/disc ratio, 0.8 (0.2–0.5); higher HOMA-IR, 7.8 (1.8–4.7). Arterio-venous nicking, a sign of vascular dysregulation, is indicated (blue arrows).

**Table 1 biology-10-00162-t001:** Clinical characteristics of a chronic stress and stroke risk phenotype cohort (N = 264).

	Stressed (N = 159)	Non-Stressed (N = 105)	*p*-Values
Age (years)	46 (±9)	45 (±9)	0.632
Ethnicity: Black	97 (61)	21 (20)	<0.001
Sex: Men	114 (72)	29 (28)	<0.001
Total energy expenditure(kcal/7 days)	3470 (±1275)	3326 (±1502)	0.405
Smoking status	45 (28)	6 (6)	<0.001
Gamma glutamyl tansferase (U/L)	68.0 (±79.2)	19.6 (13.5)	<0.001
Alcohol abuse	67 (42)	3 (3)	<0.001
Cardiometabolic risk markers			
Ethnicity-specific waist circumference	105 (66)	54 (51)	0.018
Dyslipidemia	77 (48)	34 (32)	0.011
Low grade inflammation	86 (54)	40 (38)	0.011
Cognitive_exe-func_ score	53 (±14)	62 (±13)	<0.001
Neuron-specific enolase/NSE (ng/mL)	9.9 (±4.2)	10.2 (±3.6)	0.591
S100B (ng/mL)	54.6 (±39.2)	43.7 (±20.5)	0.009
HbA1C (%)	6.1 (±0.9)	5.4 (±0.3)	<0.001
Insulin (μU/mL)	17.2 (±12)	10 (±6)	<0.001
Insulin-like growth factor 1 (ng/mL)	155 (±60)	171 (±58)	0.031
Insulin-like growth factor binding protein 3 (nmol/L)	132 (±28)	143 (±23)	0.002
Triglyceridemia	55 (35)	14 (13)	<0.001
Hyperinsulinemia	38 (24)	6 (6)	<0.001
Pre-diabetes	108 (68)	18 (17)	<0.001
Diabetes	20 (13)	0 (0)	<0.001
HOMA-IR	5 (±4)	3 (±2)	<0.001
HOMA-IR median (interquartile ranges)	4 (2, 6)	2 (1, 3)	<0.001
Upper quartile HOMA-IR (≥5)	60 (38)	11 (11)	<0.001
24-h Hypertension	118 (74)	32 (31)	<0.001
Diabetic and Hypertensive retinopathy	98 (62)	46 (44)	0.004
Retinal vascular dysregulation at 3-y follow-up			
Retinal artery (count)	12 (±2)	13 (±2)	0.046
Retinal vein (count)	11 (±1.8)	11 (±2.0)	0.949
Retinal artery caliber (MU)	1490 (±12)	153 (±12)	0.011
Retinal vein caliber (MU)	244 (±20.9)	239 (±17.2)	0.041
Focal narrowing	4 (3)	0 (0)	0.965
Arterio-venous nicking	82 (74)	29 (26)	<0.001
Optic nerve cup/disc ratio (≥0.5)	29 (19)	22 (21)	0.683
Optic nerve damage	49 (±40)	44 (±34)	0.362
Diastolic ocular perfusion pressure (mmHg)	73 (±11)	66 (±11)	<0.001
Self-reported medication			
Diabetes	8 (5)	0 (0)	<0.001
Hypertension	48 (30)	18 (17)	0.017
Anti-depressant	1 (1)	1 (1)	0.767

Values are presented as mean (±SD) or N (%). Definitions: Smoking status, cotinine, ≥14 ng/mL [36]; Alcohol abuse, GGT ≥ 41 U/L [37]; Ethnicity-specific waist circumference or central obesity cut points [44]; Dyslipidemia, total cholesterol:high-density lipoprotein cholesterol ≥5.1; Low-grade inflammation (CRP ≥ 3 ng/mL); Cognitive_exe-func_, cognitive executive functioning control; Hyperinsulinemia, insulin ≥23 μU/mL; Prediabetes, HbA1C ≥ 5.7%; Diabetes, HbA1C ≥ 6.5% [10]; Upper quartile HOMA-IR, homeostasis model assessment of insulin resistance/IR (≥5); 24-h Hypertension, SBP ≥ 130 and/or DBP ≥ 80 mmHg; Optic nerve head damage, cup-to-disc ratio ≥0.3 plus intra-ocular pressure ≥ 21 mmHg.

**Table 2 biology-10-00162-t002:** Mean changes in dementia risk markers over a 3-year period in Stressed and non-Stressed groups.

	Stressed (N = 159)			*p*-Value		non-Stressed (N = 105)		*p* = Value
	Difference (±95% CI)					Difference (±95% CI)		
Baseline Cognitive_exe-func_ score								
Baseline Telomere length (ng/μL)								
NSE (ng/mL)	+1.27 (0.52, 2.02)			≤0.001		−0.28 (−0.97, 0.41)		0.423
S100B (ng/mL)	−0.07 (−3.80, 2.46)			0.674		+2.05 (−0.27, 4.37)		0.083
WC (cm)	+3.45 (2.4, 4.6)			≤0.001		+3.40 (1.7, 5.1)		≤0.001
CRP (ng/mL)	−0.93 (−3.1, 1.1)			0.370		−1.42 (−2.5, −0.4)		0.009
VWF (%)	+16.35 (10.7, 22.0)			≤0.001		+18.94 (13.8, 24.1)		≤0.001
24-h HRV (ms)	−62.44 (170.9, 46.0)			0.257		−17.50 (−39.6, 4.6)		0.120
McNemar’s case–control test				Incidence and recovery				
HOMA-IR (Upper quartile ≥ 5)								
Incidence, n (%)			31 (25)				8 (8)	
Recovery, n (%)			3 (3)				0 (0)	
OR (95% CI), P			10.33 (3.2, 33.8), ≤0.001				To infinity	

Dependent sample *t*-test differences over 3 years (mean ± 95% CI). N = count. Abbreviations: Cognitive_exe-func_, cognitive executive functioning control; NSE, neuron-specific enolase; WC, waist circumference; CRP, C-reactive protein; VWF, von Willebrand factor; 24-h HRV, heart rate variability as determined via standard deviation of the normal-to-normal (NN) intervals between adjacent QRS complexes which equal the square root of variance. Incidence, HOMA-IR-negative at baseline becomes positive at follow-up; and where HOMA-IR-positive at baseline recovers to negative at follow-up. McNemar’s case–control test odds ratio (OR) value effects: OR 1.5 = small; OR 2.5 = medium; OR 4.25 = large.

**Table 3 biology-10-00162-t003:** Forward stepwise linear regression analyses depicting associations between retinal vessel calibers, insulin resistance and retinopathy risk markers in the Stressed and non-Stressed groups.

Model 1	Retinal Arteries (MU)	Retinal Veins (MU)
	**Stressed Group (N = 145)**	
Adjusted R^2^	0.30	0.30
	*ß* (95% CI), *p*	*ß* (95% CI), *p*
Age	-	-
Cognitive_exe-func_ score	-	−0.18 (−0.34, −0.02), *p* = 0.020
NSE (ng/mL)	-	−0.14 (−0.28, 0.00), *p* = 0.045
S100B (ng/mL)	-	0.17 (0.03, 0.31), *p* = 0.024
HOMA-IR	-	0.23 (−1.57, 2.03), *p* = 0.016
DOPP (mmHg)	−0.33 (−0.47, −0.19), *p* ≤ 0.001	0.20 (0.04, 0.36), *p* = 0.012
Optic nerve cup–disc ratio	−0.17 (−0.31, −0.03), *p* = 0.018	-
Retinal artery (count)	-	-
**Model 2**	**Non-Stressed Group (N = 96)**	
Adjusted R^2^	0.50	0.47
	*ß* (95% CI), *p*	*ß* (95% CI), *p*
Age	-	−0.23 (0.08), *p* = 0.003
Cognitive_exe-func_	-	-
NSE (ng/mL)	-	NS
S100B (ng/mL)	NS	-
HOMA-IR	-	-
DOPP (mmHg)	−0.16 (−0.34, −0.02), *p* = 0.029	-
Optic nerve cup-disc ratio	-	-
Retinal artery (count)	0.17 (0.01, 0.33), *p* = 0.027	−0.21 (−0.37, −0.05), *p* = 0.008

Case-wise deletion was applied with F to enter = 2.5. The retinal artery, neuron-specific enolase/NSE, S100B and HOMA-IR were Box–Cox transformed. Additional covariates included baseline Box–Cox transformed waist circumference, C-reactive protein and total cholesterol:high-density lipoprotein cholesterol. When the retinal artery was a dependent variable, the retinal vein was included as a covariate and vice versa. Abbreviations: Cognitive_exe-func_, cognitive executive function; DOPP, diastolic ocular perfusion pressure.

**Table 4 biology-10-00162-t004:** Logistic regression analysis indicating associations between a * validated chronic stress risk phenotype and dementia-related risk markers.

	* Chronic Stress and Stroke Risk Phenotype (N = 264)			
	Odds Ratio	5th Percentile	95th Percentile	*p*-Value
Cognitive_exe-func_	0.96	0.93	0.98	≤0.001
Telomere length	0.03	0.002	0.36	0.007
NSE (ng/mL)	0.47	0.27	0.84	0.011
Waist circumference (cm)	37.29	8.87	156.79	≤0.001
von Willebrand factor (%)	NS			
C-reactive protein (mg/L)	NS			
24-h HRV-SDNN	NS			
Mitochondrial DNA variation	0.89	0.43	1.87	0.766

All independent predictors were Box–Cox transformed, except Cognitive_exe-func_. Definitions: Cognitive_exe-func,_ cognitive executive functioning; 24-h HRV-SDNN, heart rate variability standard deviation of the normal-to-normal (NN) intervals between adjacent QRS complexes, which equal the square root of variance. * A validated chronic stress and stroke risk phenotype [14,15].

**Table 5 biology-10-00162-t005:** Logistic regression analysis to indicate associations between a Stress syndrome prototype, insulin resistance and retinopathy risk markers.

	Stress Syndrome Prototype (N = 252)			
	Odds Ratio	5th Percentile	95th Percentile	*p*-Value
HOMA-IR	7.72	2.65	22.45	≤0.001
S100B (ng/mL)	1.27	1.06	1.52	0.022
Retinal vein caliber	1.03	1.00	1.05	0.026

The Stress syndrome prototype consists of Stressed-related cognitive interference control, Box–Cox transformed telomere length, neuron-specific enolase/NSE and waist circumference. Additional covariates for the model included Box–Cox transformed markers C-reactive protein, dyslipidemia, retinal artery caliber/AV nicking, retinal artery count, optic nerve cup/disc ratio as well as carotid bifurcation stenosis and diastolic ocular perfusion pressure. OR value effects: 1.5 = small; 2.5 = medium; 4.25 = large.

## Data Availability

The data are not publicly available due to confidentiality agreement and privacy rights.

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
