# Peer review of "A Stress Syndrome Prototype Reflects Type 3 Diabetes and Ischemic Stroke Risk: The SABPA Study"

_biology, 2021, doi:10.3390/biology10020162_

Round 1

Reviewer 1 Report

The manuscript entitled "A Stress syndrome prototype reflects type 3 diabetes and ischemic stroke risk: The SABPA study" enclosed a investigation on associations between chronic stress, T3D and retinopathy risk. This is a topic worthy of study and attention. The information provided in the manuscript has a certain reference value, but it would be better to expand the sample size.

Author Response

Thank you. We agree that the sample size should be expanded. We changed the sentence in the limitations paragraph (Lines 484-486): 

“…While the prospective nature and well-controlled protocol, executed within a cohort with similar socio-economic status, are strengths of the study; our investigation was limited by the sample size and should be replicated in larger cohorts. ….”

Reviewer 2 Report

The text is very long. A concise version would increase the reader's interest.

This study indeed shares many aspects with the paper you referred to (https://www.sciencedirect.com/science/article/pii/S2666354619300286?via%3Dihub) since it is based on the same cohort. However, this study attempts to establish new statistical associations between several clinical, biochemical and physiological parameters, which was not attempted in the above-mentioned paper, thus highlighting its originality. I should also state that I have not received the similarity report you mentioned in your previous email.

Author Response

Thank you for a constructive comment. 

When we were invited to submit our paper for the special issue, it was stated in the author guidelines that the word count is not limited. However, we attempted to come up with a reasonable word count. Shortening the paper may be problematic, as the flow and logical reasoning can become compromised. 

Given the standardization of methodology and large body of work emanating from the SABPA cohort it is not surprising that some similarity with other papers is present. Compared to our previous paper [14] the similarity is 13% (please see uploaded Turn-it-in report).

Reviewer 3 Report

According to Malan and colleagues, their main goal was to “investigate associations between chronic stress, T3D and retinopathy risk”. From this statement, it is not clear which directionality their hypothesis is based on, which would be essential to improve the manuscript.

Although the authors state they have used a previously validated “chronic stress risk phenotype score, independent of age, race or gender” to stratify participants, besides the great disproportionality regarding gender, their complete data set of 264 participants (Fig. 1) showed 97 black individuals (out of 159) in the Stressed group vs only 13 black individuals (out of 105) in the non-stressed group. This could greatly indicated their analysis is biased by the highly differential proportion regarding not only gender, but also ethnicity. I would recommend the data to be comprehensively re-analysed.

In addition, there are several other concerns raised by this reviewer, which can be find below, who considered this manuscript not in conditions to be published at this point.

Material and Methods are overall very inconsistent (regarding details) and disorganised. While in some cases the authors explain it very detailed, in others they don’t say almost anything or simply refer to an article. Regarding overall methodology, all markers assessment should at least be briefly described, and then an extended methodology at supplementary data should be included. Which variables were box-cox transformed should be mentioned.

At Introduction, it should be clearly indicated which related results have been previously obtained from this same cohort / study. Ex: [15, 16], [36], [51].

A better and more extensive definition of Type 3 Diabetes should be given. Only 1 review article from 2008 is used.

The information that reflects lines 37-40 in Abstract regarding these associations should be clear from the Results obtained.

Since both NSE and S100B may represent neuronal-glia injury, and NSE remained unchanged between groups (Table 1), why would the authors claim that “The Stressed group had (…) consistent neuronal-glia injury”, since that is not exactly what is seen from Results?

Regarding Table 1:

There are some variables (with statistically differences between groups) that should be previously mentioned in the Results text, and not only then referred in Discussion (alcohol abuse, triglyceridemia). Moreover, other such as the smoking status, or the measurement of GCT, were never mentioned.

It is not clear whether “Ethnic-specific central obesity” refers to WC or how it is related to, as comparing with Figure 2a the results are quite different.

It is surprising HOMA-IR having p<0,001 with such high SD.

How “Focal narrowing” was obtained is not shown.

It is not clear why, for the same variable, there are different values in Figure 1 vs Table 1. Ex: ethnicity non-stressed (4 men + 9 women vs 21), which makes 80% white + ~13% black ≠ 100%!

Regarding Figure 2:

Even if different units were used for obtaining the bars, an y axis global legend should be included or, preferably, individual graphs should be shown.

It is surprising Telomere length having p<0,001 with such high SE.

The authors should better explain the minimal (but observed) differences in values for HOMA-IR, NSE and S100B shown in graphs when compared to Table 1. Importantly, while no statistically differences were obtained for NSE in Table 1 (9.9 vs 10.2; p=0,591) a p<0.001 is denoted in Figure 2b (10 vs 11), which is surprising.

Statement in lines 283-4 was never observed in Figures or Tables.

To validate the statement in lines 292-3, Table 1 or Figure 2 should be also referred.

Regarding Table 4, exactly how and why were these markers chosen, should be stated and clearly explained.

I would suggest to decrease size of Figure 3 and move Figure 4 to Supplementary data.

In addition:

At methods, the authors claim evaluation of “Dementia risk markers” which included HOMA-IR, neuronal-glia injury, telomere length, cognitive/ex/func, central obesity or WC, VWF, CRP, HRV-SDNN, but don’t mention how or when, only showing references for previous articles. The authors should fully justify their specificity for Dementia (at Introduction or Results).

Then, the authors claim evaluation of “Cardiovascular measurements” which included BP / ECG, retinal vessel imaging and anthropometric measures, but then also refer the Stroop task (= cognitive/ex/func), and still it is also where they describe the ambulatory and in-patient procedure, so absolutely not an adequate title.

Moreover, in contrast to previous, a “Fundus imaging and Retinopathy” section very deeply describes the methodology for retinal vessel imaging, which had been already referred in “Cardiovascular measurements”.

A section of “Cognitive executive functioning control” describes the use of the “Stroop test”, previously named “task” (what is the difference of test vs task?), previously referenced in the first 2 methods sections. Here, inconsistently to previous, extensive background is given (5 different references are used!). While previously stating 8 different markers as “Dementia markers” (section 1 from Methods), here they state it was used “as dementia risk marker” (would this mean the others are not specific?), although the test has “also been defined as generic defense response”. How do the authors differentiate these? Finally, the test included financial encouragements. Can the authors prove that this did not cause bias on the results, as not every individual might respond with the same effort to this kind of motivation?

Then, “Lifestyle risk markers” included WC and Mean total energy expenditure, both of which don’t exactly or individually reflect lifestyle. Perhaps the authors definition of lifestyle risk should be disclosed. Interestingly, variables such as body weight, BMI, etc, seem to be missing. Authors should better explain why WC was previously considered Dementia risk marker.

In fasted “Biochemical analyses”, it includes glucose (which apparently was only used to calculate HOMA-IR), lipids (which need to be discriminated), cotinine (which should at least mention that it is used to evaluate the smoking status), CRP (previously considered as Dementia marker, which should be better discussed), IGF1 and IGFBP3 (but not indicated why these were measured neither why lower levels - and at which threshold - were considered as diabetes-prone symptoms), HbA1c (used to define pre- and diabetes status), HOMA-IR (but did not mention that - neither how - insulin levels were assessed), VWF (marker of ischemic stroke / coagulation activation), NSE, S100B, Mitochondrial DNA variation (to establish an association between stress and maternal lineage; however, this hypothesis was never clearly mentioned and only a brief sentence was used in line 329-30 to state the negative results obtained; in which sample should also be stated), Leucocyte telomere length genomic DNA (although showed in Supplementary data, a brief note of the method should be given in main text), and HIV positive status (which was the only moment it was referred in the manuscript, so it is not clear the relevance for this analysis if then it is not included in the analysis).

Finally:

The results summarised in Discussion (lines 366-372) should be better specified in the Results section, such as the exact T3D signs mentioned.

References numbering should be revised. Ex: line 69 references S100B with [12], which refers to NSE, whereas then NSE is referenced with [13], which seems to refer to S100B; Ref [49] does not seem to be associated to central obesity; Ref [50] does not seem to be related to people of African descent, but [52] does.

In Table 1, HbA1c, IGF1, IGFBP3 means lack +- for SD in ().

Round 2

Reviewer 3 Report

I appreciate the authors' effort and care in responding to all my concerns. I believe the manuscript might now be considered to be published.